# Cation Permeability of Voltage-Gated Hair Cell Ca^2+^ Channels of the Vertebrate Labyrinth

**DOI:** 10.3390/ijms23073786

**Published:** 2022-03-29

**Authors:** Marta Martini, Giorgio Rispoli

**Affiliations:** Department of Neuroscience and Rehabilitation, Section of Physiology, University of Ferrara, Via Borsari 46, 44121 Ferrara, Italy; mrm@unife.it

**Keywords:** patch clamp, semi-circular canals, Ca^2+^ channel selectivity, Ca^2+^ current, nifedipine, ion channels, voltage-sensitive channels

## Abstract

Some hearing, vestibular, and vision disorders are imputable to voltage-gated Ca^2+^ channels of the sensory cells. These channels convey a large Ca^2+^ influx despite extracellular Na^+^ being 70-fold more concentrated than Ca^2+^; such high selectivity is lost in low Ca^2+^, and Na^+^ can permeate. Since the permeation properties and molecular identity of sensory Ca^2+^ channels are debated, in this paper, we examine the Na^+^ current flowing through the L- and R-type Ca^2+^ channels of labyrinth hair cells. Ion currents and cytosolic free Ca^2+^ concentrations were simultaneously monitored in whole-cell recording synchronous to fast fluorescence imaging. L-type and R-type channels were present with different densities at selected sites. In 10 nM Ca^2+^, the activation and deactivation time constants of the L-type Na^+^ current were accelerated and its maximal amplitude increased by 6-fold compared to physiological Ca^2+^. The deactivation of the R-type Na^+^ current was not accelerated, and its current amplitude increased by 2.3-fold in low Ca^2+^; moreover, it was partially blocked by nifedipine in a voltage- and time-dependent manner. In conclusion, L channel gating is affected by the ion species permeating the channel, and its selectivity filter binds Ca^2+^ more strongly than that of R channel; furthermore, external Ca^2+^ prevents nifedipine from perturbing the R selectivity filter.

## 1. Introduction

Ca^2+^ influx through voltage-gated Ca^2+^ channels is the first step in synaptic transmission; in vestibular system hair cells, these channels sustain transmitter release at the ribbon synapse, located at the cell basal pole [1,2,3,4]. The ribbons endowing this peculiar synapse, typical of hair cells and retinal photoreceptors, seem to be designed to support continuous exocytosis by speeding up the delivery of vesicles to release sites, by promoting molecular priming and synchronous exocytosis of several vesicles, but preventing depletion or unneeded release in inactive synapses (reviewed in [4]).

Ca^2+^ channels are classified according to the properties of the current they carry: the L-type currents originate from the Ca_v_1 channel family; the P/Q-, N-, and R-type currents arise from the Ca_v_2 family; and T-type currents are mediated by the Ca_v_3 family. This classification is the outcome of immense interdisciplinary research lasting more than fifty years, primarily studying the effects of specific drugs on these channels in ex vivo tissues, in vivo, and expressed (native or mutated) in cell lines [5]. A large number of genetic diseases are linked to mutated Ca^2+^ channels and in their modulatory proteins, and particularly in some hearing, vestibular, and vision disorders that are probably imputable to Ca^2+^ channels at the level of the sensory cells. For instance, mutations in the subunits of the Ca_V_1.3 channel type and in its modulator protein CaBP2 originate the autosomal recessive deafness, while many mutations in Ca_V_1.4 channels and in their modulator protein CaBP4 are associated with autosomal recessive congenital stationary night blindness (reviewed in [4]). Vestibular disorders such as type 2 episodic ataxia are caused by mutations in the P/Q-type channel [6], possibly at the level of the vestibular hair cells [7]. Therefore, it is of paramount importance to investigate the biophysical properties of these channels: a convenient model is the hair cells isolated from the rana esculenta labyrinth, because they are particularly large and robust, yet they are very similar to the other vertebrate hair cells, and this comestible species can be easily supplied by local dealers.

Voltage-gated Ca^2+^ channels are extremely selective for Ca^2+^ ions, since they convey a Ca^2+^ influx of 10^6^ ions/sec into the cell, although extracellular Na^+^ is 70-fold more concentrated than Ca^2+^ and both ions have nearly identical dimension. Indeed, the van der Waals, ionic, and hydrated radius of Ca^2+^ are 2.31, 1.00, and 4.12 Å, respectively, while those of Na^+^ are 2.27, 1.17, and 3.58 Å [8], respectively, suggesting that the Ca^2+^ channel selectivity filter could rely on recognising the hydrated form of Ca^2+^ to select it over Na^+^ (see Section 3 Discussion).

It has long been known that this high selectivity is lost if the extracellular Ca^2+^ concentration is lowered from the physiological concentration (few mM) to 1–10 µM, and other mono- and divalent cations can permeate [9,10]. More recently, Ca^2+^ channel permeability to monovalent cations has been investigated in turtle cochlear hair cells [11], which appear to display an L-type Ca^2+^ channel [12]. It is still not clear what Ca^2+^ channel types are endowing the different sensory cells, and in general, it is important to understand the permeation properties of different channel types. To this aim, the hair cells of the frog semi-circular canal are particularly significant, since three distinct Ca^2+^ channel types have been identified in these cells on the basis of their biophysical and pharmacological properties [2]. In these cells, a substantial fraction of the Ca^2+^ current flows through a nifedipine-sensitive L-type channel; the remaining fraction (after nifedipine application at saturating concentration) flows through two distinct R-type channels, termed R1 and R2 for simplicity’s sake, on the basis of the following evidence. It has been found that approximately 60% of the hair cells recorded in whole-cell mode exhibited a steady current, maximal at −20 mV, whereas the remaining 40% were characterised by an initial peak, followed by an exponential decay (inactivation) to a plateau level. Peak amplitude was maximal at −30 mV, and inactivation was Ca^2+^-dependent [2], like other R-type channels described in neurons [13]. Nifedipine (or nimodipine) concentrations of 1, 5, and 10 μM gave the same fractional block of the total current (about 60–70%; [2,14]), without affecting the peak phase of the current ([2]; see Section 2 Results). This indicates that 60–70% of the total current flows through an L-type channel (Ca_v_1), and the current formed by the peak phase and the residual (steady) current flows through two distinct R-type channels (Ca_v_2), called R1 and R2 hereafter, on the basis of their resistance to dihydropyridines, ω-cono- and aga-toxins (thus ruling out the presence of N- or P/Q-type Ca^2+^ channels; [2]). No correlation was found between the peak and the plateau amplitudes; for example, the peak amplitude was progressively reduced, and the inactivation time constant progressively slowed down, with repetitive depolarisations (run-down), without any appreciable change in the plateau amplitude or in the residual current fraction following nifedipine blockade of the L component [2]. This evidence excludes that the peak followed by the inactivation phase was part of the L or of the R2 components.

Clear evidence for L-type Ca^2+^ channel has been found in hair cells of the turtle cochlea [12] and in bullfrog sacculus [14,15,16]. The Ca_v_1.3 channel types have been found in chicken cochlear hair cell [17] and in vestibular epithelia of the rat, but Ca_v_1.2 types have been found in the latter vestibular ganglia [18]. The hair cell L-type Ca^2+^ channels differs from the other known L-type Ca^2+^ channels (Ca_v_1.1, Ca_v_1.2, Ca_v_1.3, and Ca_v_1.4) in that they are activated by low rather than high voltages [19], suggesting that the low-voltage activation could be conferred by some post-transcriptional modification and/or differential association with auxiliary subunits [20] and/or some channel-associated cytoplasmic protein(s) [21]. It has been shown [22,23] that the Ca_v_1.3 channels expressed in cell lines actually form low-voltage activated channels with low nifedipine sensitivity, resembling the hair cells L-type Ca^2+^ channels, which indeed require micromolar amounts of dihydropyridine to block them, instead of nanomolar amounts as in other L-type channels. Remarkably, bullfrog saccular hair cells possess two Ca^2+^ channels: L- and non-L-type [14,15,16]. Evidence for non-L Ca^2+^ currents in mammals is indicated in the chinchilla crista by labelling of the basolateral membrane in type I and type II hair cells with an antibody against the subunits of P/Q channels (Ca_v_2.1; [24]); moreover, an R-type (Ca_v_2.3) channel has been found in the hair cell membrane in the mouse organ of Corti [25]. The presence of a non-L Ca^2+^ current component is also supported by the lack of vestibular deficits in the Ca_v_1.3 null mice [26]. Nimodipine-insensitive Ca^2+^ currents to be mediated via Ca_v_2 channels were found in bullfrog saccular hair cells [15], and recent results have conclusively demonstrated that the R-type Ca_v_2.3 Ca^2+^ channels are mandatory for physiological auditory information processing [27]. Therefore, it is likely that R-type channels are present in hair cells of semi-circular canals as well; however, no molecular biology studies have conclusively demonstrated their presence, together with the well-established L-type channel. Moreover, the R1 component, originally identified as an R-type because it was sensitive to the R-type channel blocker mibefradil [2], could be instead a T-type channel that is mibefradil-sensitive as well [28], and possibly a Ca_v_3.1 one, given its fast inactivation kinetics [29]. In conclusion, the channel classification presented here should be considered still not conclusive.

The L and R2 currents, which activate at −60 mV and peak at −20 mV, may sustain spontaneous transmitter release at the cytoneural junction, whereas the R1 component, which activates at −40 mV and carries the maximal current at −30 mV, may be functionally important in evoking synchronous transmitter release in response to short, strong stimuli [2]. Given the still scarce information on the permeation properties of hair cell Ca^2+^ channels and on their molecular identity, the present work examines the non-specific monovalent cation current flowing through the L- and putative R-type Ca^2+^ channels in hair cells of the frog labyrinth in the presence of very low external Ca^2+^, by focusing on their (i) activation and deactivation kinetics, (ii) voltage-dependent conductance properties, and (iii) selectivity and blockade. The results offer a preliminary step in understanding the basic properties and structure of Ca^2+^ channels in such specialised sensory cells. They will also be helpful in analysing the permeation properties of functionally different channel types that share the same ion carrier.

## 2. Results

### 2.1. Isolation of L- and Putative R-Type Components

As found previously, approximately 60% of the hair cells recorded in whole-cell mode exhibited a steady current, maximal at −20 mV, whereas the remaining 40% were characterised by an initial peak, followed by an exponential decay (inactivation) to a plateau level. Peak amplitude was maximal at −30 mV, and inactivation was Ca^2+^-dependent [2]. No correlation was found between the peak and the plateau amplitudes; for example, if the plateau amplitude was reduced by 70% with nifedipine, the peak component amplitude was negligibly affected, and its inactivation (average inactivation time constant 7.2 ± 1.0 ms, *n* = 9) was not slowed down by the massively reduced Ca^2+^ influx (see below), as indeed occurred when external Ca^2+^ was lowered [2]. This indicates that the channels are clustered in microdomains, and therefore, the inactivation of an R1 channel is little influenced by other channel types nearby. All three channel types (R1, R2, and L) were strongly selective to Ca^2+^ in the presence of a physiological level of Ca^2+^, and no Na^+^ was flowing through any of these channels, as found instead in other Ca^2+^ channel types, such as the Ca_v_3 ones (T-type; [30]). Indeed, replacing all external 100 mM Na^+^ with impermeant cation choline, in the presence of invariant 4 mM Ca^2+^, produced no significant changes in current amplitude or waveform elicited by the depolarisation in a hair cell (Figure 1A) that contained all three channel types (Figure 1B). This also excludes the presence of Na_v_ channels, such as the TTX insensitive ones that peak at −30 mV, recently described in human foetal vestibular hair cells [31]; otherwise, the current amplitude (in response to a −30 mV depolarisation; Figure 1B) would be reduced in choline. This experiment also excludes the presence of a TTX-sensitive Na_v_ channel, such as the one described in type I and type II auditory hair cells of the chicken (blocked by sub-micromolar concentrations of TTX; *K_d_* ≈ 17 Nm [32]). Other evidence demonstrating the absence of a TTX-sensitive Na_v_ is the almost identical current waveform, elicited by depolarisations in 4 mM or in 10 nM external Ca^2+^, recorded before, in the presence, and after the application of TTX (1 μM, using the fast perfusion system described in Section 4 Methods and shown in the video microphotograph of Figure 2). Finally, there was no significant change in the spatial distribution (Figure 1C) amplitude or kinetics of the Ca^2+^ dynamics (Figure 1D) elicited by a 160 ms depolarisation to −30 mV that revealed Ca^2+^ entry at selected sites (hotspots) located mostly in the basal (synaptic) pole of the cell, where the Ca^2+^ channels are indeed clustered (Figure 1C). In Figure 1C, just one hotspot is visible, but the spatial resolution of these experiments, being non-confocal, do not allow us to distinguish hotspots located, for instance, at two ends of the cell but along the same optical axes, since the camera integrates the fluorescence signal along this axis; nevertheless, it was possible, in many other cells, to discriminate several hotspot (see below; [33]).

### 2.2. Current Elicited by Depolarisation in 10 nM Ca*^2+^*

Upon reducing external Ca^2+^ to 10 nM, a large current was recorded in response to a depolarisation to −50 mV (Figure 2); no current was recorded in 10 nM Ca^2+^ when external Na^+^ was replaced with an equimolar concentration of choline. This demonstrates that the current flowing through the Ca^2+^ channels is entirely carried by Na^+^ at −50 mV, since at negative voltages, no Cs^+^ efflux though the channels is expected, its equilibrium potential being ~−70 mV. However, at more depolarised voltages, it is possible that Cs^+^ carries a significant fraction of the total current; therefore, in the following, the current flowing through the Ca^2+^ channels in the presence of 10 nM external Ca^2+^ will be referred to as the “cation current”. With the perspective that the channel classification as R1, R2, and L is not conclusive, it is useful to examine the permeation properties of the cells possessing the L and putative R2 channels separately from those possessing all three channel types. Cells possessing L + R2 components only displayed steady current at all voltages in both 4 mM (Figure 2A) and 10 nM (Figure 2B) external Ca^2+^.

The steady-state current to voltage (I–V) curves (Figure 2C) of the Ca^2+^ (Figure 2A) and of the cation (Figure 2B) currents, as well as their normalised average I–V curves (Figure 2D, *n* = 21), were maximal at −20 mV in 4 mM Ca^2+^ (Figure 2C, filled circles) and between −50 and −60 mV in 10 nM Ca^2+^ (Figure 2C, filled triangles). The normalised averaged I–V curve profile of the cation current was very similar to the Ca^2+^ current one, but shifted by ~30 mV leftwards (Figure 2D); the ratio between the two average maximal currents was 4.77 ± 0.76 (average current in 4 mM Ca^2+^: 95.8 ± 7.8 pA, *n* = 21; average current in 10 nM Ca^2+^: 457 ± 36 pA, *n* = 21, see also below; all statistical results are reported in Table 1). Different cells had obviously different current amplitudes elicited by the same depolarisation depending on their dimension, their Ca^2+^ channel number, etc., but for all recorded cells, Ca^2+^ current amplitudes were linearly related (*r* = 0.90, *n* = 21) to the cation currents at the same voltage minus ~30 mV. These results confirm once more that all the recorded currents were carried by monovalent cations flowing through the open Ca^2+^ channels. Moreover, the ~−30 mV shift was then mainly due to the increased net negative charge on the external side of the plasma membrane produced by the removal of Ca^2+^ positive charges adsorbed there.

Cells possessing all three components (L + R1 + R2) responded to the depolarisation step with a characteristic Ca^2+^-dependent inactivation waveform, being maximal for a −30 mV depolarisation (Figure 2E). Most of these experiments were carried out in the perforated-patch configuration because this technique delayed the run-down of the R1 component, which faded away quickly during whole-cell recordings [2]. Currents recorded in perforated patch were, however, approximately three times larger than those recorded in whole-cell configuration; consequently, in 10 nM Ca^2+^, the currents were sometimes so large that they hindered reliable cell voltage clamping. Therefore, since it was not possible to compensate access resistance by more than 60% (larger compensations generated patch amplifier oscillations), cells with perforated-patch currents and access resistance no larger than 1.6 nA and 10 MΩ, respectively, were considered. The depolarisations elicited steady currents in 10 nM Ca^2+^, because the R1 component did not inactivate. Consequently, it was not possible to distinguish the R1 from the R2 component in the low Ca^2+^ recordings, or to directly assess whether R1 was still present or if it had undergone run-down. Therefore, at the end of the 10 nM Ca^2+^ perfusion, the presence of R1 was once more checked by returning the cell to the 4 mM Ca^2+^ solution and repeating the depolarisation protocol of Figure 2E. R1 run-down was also minimised by shortening the length of the experiment, stimulating the cell with only one depolarising step in 4 mM Ca^2+^ (Figure 2E) and with only four in 10 nM Ca^2+^ (Figure 2F; the complete I–Vs of the Ca^2+^ current and of the cation current are shown below, recorded from particularly stable cells). The steady-state I–V curve (Figure 2G) and the normalised average curve (Figure 2H) of the cation current in 10 nM Ca^2+^ (Figure 2F) were maximal at ~−50 mV; the cation current was 2.95 ± 0.89 and 3.45 ± 1.02 (*n* = 9) times larger than the peak and the plateau Ca^2+^ currents in 4 mM Ca^2+^, respectively (Table 1). Once again, there was a linear relationship (*r* = 0.84, *n* = 9) between different peak current amplitudes in 4 mM Ca^2+^ (at −30 mV) and the corresponding maximal cation current amplitudes in 10 nM Ca^2+^ (at −50 mV) in different cells, showing that the cation current flowed also through the R1 channels.

### 2.3. L-Type Channel Blockers

In order to further determine the contribution of each channel type to the generation of current responses in 10 nM Ca^2+^, experiments were designed to block the major current component: the L-type. Since nifedipine concentrations between 1 and 10 μM gave the same suppression of the Ca^2+^ current, it was concluded that this concentration range sufficed to fully block the L component of the current. In the following, concentrations of 10 μM were chosen to ensure the full block of the L component, but no larger concentrations were employed, since they block other Ca^2+^ channel types [34] and could therefore affect the R1 and R2 components of the current as well. Ca^2+^ current amplitude and kinetics evoked by repeated depolarisations during a 4 min perfusion with the L channel blocker calciseptine (2 µM; *n* = 3), specific for Ca_v_1.1 channels in skeletal muscle [35], and calcicludine (20–100 nM; *n* = 4) specific for Ca_v_1.2 channels [36], were not affected (data not shown). Therefore, these compounds were not blockers (or agonists) of the open or closed state of any of the vestibular hair cell channels. Since nifedipine concentrations between 1 and 10 µM blocked about the same fraction of total Ca^2+^ current [2,14], it was concluded that 10 µM nifedipine was enough to completely remove the L-type contribution.

To better understand the contribution of L, R1, and R2 contributions to the current waveform and the properties of the cation current flowing through these channels in 10 nM Ca^2+^, we examined first the current kinetics of cells having L + R2 components only and when the L component was removed with 10 µm of nifedipine.

The activation kinetics of maximal Ca^2+^ currents (at −20 mV) in cells having L + R2 components only could be conveniently fitted by a single exponential, as exemplified in Figure 3A, even though they were generated by the activation of two channel types (*τ**_on_* = 0.70 ± 0.06 ms, *n* = 21), while the deactivation phase (Figure 3A) was bi-exponential (*τ**_off1_* = 1.18 ± 0.16 ms, *τ**_off2_* = 9.1 ± 1.1 ms, *n* = 21). Moreover, the *τ**_on_* in nifedipine (0.75 ± 0.11 ms; *n* = 11; Figure 3B) was similar to the one in the control, but the deactivation kinetics were monoexponential, with a time constant (5.8 ± 0.5 ms; *n* = 11) comparable to the *τ**_off2_* in the control. This indicates that nifedipine indeed completely cancelled the L component of the current, whose activation and deactivation time constants were similar and faster than the R2 (which is then isolated in Figure 3B) ones, respectively.

The *τ**_on_* of the maximal cation current (at −50 mV; 0.36 ± 0.02 ms, *n* = 21) in cells possessing the L and R2 components only (Figure 3C) was about half of the Ca^2+^ current one. This reduction of *τ**_on_* was not due to the different voltages at which the maximal current was evoked (−20 mV in 4 mM Ca^2+^ and −50 mV in 10 nM Ca^2+^) because *τ**_on_* was relatively voltage-independent, and it even increased with hyperpolarisation, as found for T-type channels [30]; the voltage dependence of *τ**_on_* in 10 nM Ca^2+^ was similar to the one in 4 mM Ca^2+^, but shifted by ~−30 mV. The deactivation phase of the maximal cation current was bi-exponential as the Ca^2+^ current ones, but *τ**_off1_* (0.36 ± 0.02 ms; *n* = 21) and *τ**_off2_* (12.0 ± 1.8 ms; *n* = 21) were three times faster and similar to the ones of the maximal Ca^2+^ current, respectively. The activation kinetics of the maximal cation current (at −50 mV) in nifedipine (which flows through the isolated R2 channels) was fitted by a single exponential (Figure 3D), with a *τ**_on_* (0.38 ± 0.04 ms, *n* = 8) twice as fast as that found in 4 mM Ca^2+^ + nifedipine (at −20 mV). In 10 nM Ca^2+^, the deactivation kinetics in nifedipine were monoexponential as in 4 mM Ca^2+^, with a similar time constant (6.3 ± 1.0 ms, *n* = 8). All these results indicate that 10 nM Ca^2+^ accelerated the activation time constant of both L and R2 components by a factor of ~2, but accelerated the deactivation time constant of the L component (by a factor of ~3) only.

The *τ**_on_* of the maximal Ca^2+^ current generated by the activation of L + R1 + R2 channel types (0.39 ± 0.12 ms; *n* = 9; Figure 1B and Figure 2E; see also below) was about twice as fast as the *τ**_on_* of L + R2 Ca^2+^ current. This is probably due to the underestimation of *τ**_on_* of the L + R1 + R2 Ca^2+^ current, since the inactivation phase of the R1 component cut in before the activation phase of the entire current (through the L + R1 + R2 channel types) had fully developed, thus artificially shortening it. The deactivation phase was bi-exponential (*τ**_off1_* = 0.89 ± 0.19 ms, *τ**_off2_* = 10.6 ± 1.0 ms, *n* = 9), and as in cells possessing the L and R2 components only, the *τ**_on_* in nifedipine (0.40 ± 0.04 ms; *n* = 9) was similar to the one in the control, but the deactivation kinetics was monoexponential, with a time constant (6.6 ± 1.6 ms; *n* = 9) comparable to the *τ**_ff2_* in the control. These results are consistent with the notion that the R1 component fully inactivates during the depolarisation, and the current deactivation was generated by the sum of the fast-deactivating L component and the slow-deactivating R2 component.

The *τ**_on_* of the L + R1 + R2 maximal cation current (0.37 ± 0.04 ms; *n* = 9; Figure 2F; see also below) in 10 nM Ca^2+^ was similar to the *τ**_on_* in 4 mM Ca^2+^, but the latter data were unreliable, as explained above. Therefore, it cannot be assessed whether in 10 nM Ca^2+^ the *τ**_on_* of the R1 component was accelerated (as the *τ**_on_* of the L and R2 components described above) or not. Since the R1 component did not inactivate in 10 nM Ca^2+^, the deactivation kinetics of the cation current were generated by the deactivation of L, R1, and R2 components. The latter was still biexponential (i.e., as in L + R2 cells): the cation current *τ**_off1_* (0.29 ± 0.02 ms; *n* = 9) was about threefold faster than that of the Ca^2+^ current, while *τ**_off2_* was similar for both currents (11.6 ± 1.9 ms; *n* = 9); the *τ**_off1_* and *τ**_off2_* of cation current were similar to the ones of L + R2 cells. If *τ**_off2_* is generated by the R2 inactivation only (as discussed above for cells with R2 and L components only), then *τ**_off1_* is generated by the inactivation of L and R1 components, and since this inactivation was threefold smaller in 10 nM Ca^2+^, then the inactivation of both R1 and L components was accelerated in 10 nM Ca^2+^. This view is confirmed by perfusing with nifedipine the L + R1 + R2 cells: after cancelling the L component, the *τ**_on_* (0.28 ± 0.04 ms; *n* = 6) was accelerated compared to 4 mM Ca^2+^, and the deactivation was biexponential (*τ**_off1_* = 0.32 ± 0.08 ms; *τ**_off2_* = 5.8 ± 1.7 ms; *n* = 6) instead of monoexponential as in 4 mM Ca^2+^. These results indicate that, after cancelling the L-type component in nifedipine in 10 nM Ca^2+^, the biexponential kinetics of monovalent ion current deactivation are described by a *τ**_off1_*, generated by the deactivation of the R1 component, and by *τ**_off2_*, generated by the R2 component. It cannot, however, be ascertained if the deactivation of the R1 component was accelerated or not in 10 nM Ca^2+^.

The L, R1, and R2 channels are clustered in hotspots located mostly in the basal (synaptic) pole of the cell (Figure 1C and Figure 3F, middle and lower panels). In experiments, currents were subjected to run-up or run-down in cells exhibiting clearly distinct hotspots; the fluorescence did not uniformly decrease or increase at all hotspots [33]. This indicates that intracellular signalling mechanisms regulate differently distinct Ca^2+^ channel types, distributed at different densities in distinct hotspots. This view is consistent with the experiments of Figure 3F, where two cells chosen for having different morphologies and several hotspots and exhibiting all three channel types were extracellularly perfused with nifedipine. The blockade of L-type components reduced the fluorescence (Figure 3F, two images) differently at different hotspots compared to the control (middle two images), indicating that the L-type channels have different densities at different hotspots. In the experiments, the current amplitude and waveform (not shown, but similar to the one of Figure 1B) did not change before and after nifedipine application.

### 2.4. I–Vs of L + R2 and of L + R1 + R2 Currents in 4 mM Ca*^2+^*

In cells possessing the L and R2 components only (Figure 4A), 10 µM nifedipine reduced maximal current (at −20 mV) in 4 mM Ca^2+^ by a factor of 2.43 ± 0.45 (average current in the control, i.e., in 0 nifedipine: 95.8 ± 7.8 pA, *n* = 21; average current in 10 µM nifedipine: 39.4.8 ± 4.1 pA, *n* = 11; Figure 4B). Therefore, ~60% of the total current was flowing through the L-type channels, while the remaining ~40% was flowing through the R2 channels. The I–V in the control (Figure 4C, circles) of the isolated R2 current (Figure 4C, diamonds) and of the isolated L-type current (obtained by subtracting the former I–V from the latter one; Figure 4C, squares turquoise) have similar trends, indicating that the L-type channel blockade was negligibly voltage-dependent.

In cells possessing the L, R1, and R2 components (Figure 5A), 10 µM nifedipine reduced the peak and the plateau amplitude of the maximal current (at −30 mV) by about 3.4-fold (average current in the control: peak: 421 ± 80 pA, plateau: 359 ± 66 pA, *n* = 9, Figure 5A; average current in 10 µM nifedipine: peak: 144 ± 31 pA, plateau: 95 ± 22 pA, *n* = 9, Figure 5B). Therefore, ~70% of the total plateau current, once the R1 current was completely inactivated, was flowing through the L-type channels, while the residual ~30% was flowing through the R2 channels. This indicates that the L-type component was somewhat larger in the cells with all three channel types than in those with the L and R2 channels only. For a better view of the nifedipine effect on the current waveform, selected traces to the same depolarisation in the control (Figure 5A) and in nifedipine (Figure 5B) are shown in colour in Figure 5C. The difference (red-filled circles, Figure 5D) between the I–V of the peak (open circles) and the I–V of the plateau (black-filled circles) of the current in the control was similar to the difference (green-filled circles) between the I–V of the peak in nifedipine (open diamonds) and the I–V of its plateau (black-filled diamonds): both differences represent the I–V of the R1 component. The difference between the plateau I–V in the control and the one in nifedipine (cyan-filled squares) gives the I–V of the L component, whose shape is similar to the one in Figure 4C.

### 2.5. I–Vs of L + R2 and of L + R1 + R2 Currents in 10 nM Ca*^2+^*

In cells exhibiting the L and R2 components only (Figure 6A), the monovalent ion current recorded in 10 nM Ca^2+^ (Figure 6B) was not simply reduced by nifedipine (Figure 6C) by a constant scaling factor as was found above in 4 mM Ca^2+^ (Figure 4). Surprisingly, in nifedipine, the isolated R2 component decayed exponentially during the depolarising steps (Figure 6C; time constant: 6.7 ± 0.7 ms at −50 mV; *n* = 8): nifedipine reduced the maximal cation current (at −50 mV) by a factor of 2.93 ± 0.49, which became 4.57 ± 0.39 at the plateau (average current in 10 nM Ca^2+^: 457 ± 36 pA; average current in 10 nM Ca^2+^ + 10 µM nifedipine: peak: 156 ± 14 pA, plateau: 100 ± 8.0 pA, *n* = 8). Moreover, the nifedipine cation current blockade was stronger at more negative potentials (Figure 6D,E).

In cells exhibiting all three components (Figure 7A), the cation current in 10 nM Ca^2+^ (Figure 7B) was blocked by nifedipine (Figure 7C) in a time- (average time constant: 6.2 ± 1.01 ms at −50 mV; *n* = 6) and voltage-dependent manner (Figure 7D,E), consistently with the results shown in Figure 6. Nifedipine reduced the maximal cation current (at −50 mV) by a factor of 3.91 ± 1.32, which became 5.99 ± 2.47 at the plateau (average current in 10 nM Ca^2+^: 1250 ± 140 pA, *n* = 9; average current in 10 nM Ca^2+^ + 10 µM nifedipine: peak: 317 ± 71 pA, plateau: 207 ± 62 pA, *n* = 6; Figure 7B,C). This block was stronger compared to the one observed in cells lacking the R1 component; since the latter do not inactivate in 10 nM Ca^2+^, this larger block is possibly due to a partial block of the R1 component as well when cations carry the current. In any case, the time-dependent block was clearly observed for voltages between −10 and −60 mV, i.e., when the inward current had the largest values (Figure 6C and Figure 7C).

## 3. Discussion

With the aim of better identifying the permeation properties of the R1, R2, and L channel types in frog semi-circular canal, experiments were designed to discern the behaviour of each one by analysing their Ca^2+^ and cation current waveforms. This was accomplished by confronting the Ca^2+^ and the cation currents generated by cells possessing all three channel types with these generated by the cells containing solely the L and R2 channels, and by blocking the L-type component with nifedipine. Regarding the L-type channel, on the basis of the extensive literature discussed in the introduction, and the lack of any observed effect of calciseptine and calcicludine (thus ruling out the presence of the Ca_v_1.1- and Ca_v_1.2-type subunits; [37]), it can be safely concluded that it is formed by the Ca_v_1.3 subunit.

Voltage-gated Ca^2+^ channels are very similar to sodium channels in structure and function, yet subtle differences in their amino acid sequences allow them to convey a Ca^2+^ influx of 10^6^ ions/sec into the cell, despite extracellular Na^+^ being 70-fold more concentrated than Ca^2+^. In fact, Ca^2+^ channels have a Ca^2+^ selectivity much larger than the Na^+^ one, especially the ones analysed here, whose Na^+^ inflow in the presence of physiological external Ca^2+^ is not measurable (Figure 1). This is extremely demanding, since Ca^2+^ and Na^+^ have nearly identical van der Waals and ionic radii; moreover, when external Ca^2+^ was reduced to <10 nM, monovalent cations permeated every Ca^2+^ channel type, since Na^+^ removal cancelled out the total current, no matter what set of channel types was present in the recordings. The Na^+^ current increased 5–6-fold over the Ca^2+^ current; Na^+^ was not able to substitute for Ca^2+^ at the R1 channel inactivating site. How Ca^2+^ channels solve this fundamental biophysical problem has been investigated for almost four decades [9,10], but only recently, a satisfactorily explanation, yet still debated, was proposed [38,39,40]. This phenomenology has been classically explained by assuming that the channel is endowed with at least one binding site with higher affinity for Ca^2+^ than for monovalent cations (as Na^+^). In the absence of Ca^2+^, monovalent cations loosely occupy this binding site and therefore can flow relatively freely through the channel, along their electrochemical gradient. As the Ca^2+^ concentration is raised, this binding site becomes occupied by Ca^2+^, since monovalent cations are not able to displace it, given the higher affinity of the site for Ca^2+^ than for Na^+^; consequently, the bound Ca^2+^ actually blocks current flux through the channel. As the concentration of Ca^2+^ is raised even further, multiple binding sites in the pore become occupied by Ca^2+^; the electrostatic repulsion between the multiple bound Ca^2+^ ions would provide the driving force to move Ca^2+^ through the pore (often referred to in the literature as the knock-off mechanism), despite its tight binding [9,10], as demonstrated for the K^+^ channels [41]. More recently, on the basis of electrophysiological and crystallographic analysis of a Ca^2+^ selectivity filter constructed in a homotetrameric bacterial Na^+^ channel, it has been proposed that this filter is composed of three binding sites (named, for the sake of simplicity, Site 1, Site 2, and Site 3; [38]). Site 1, positioned close to the extracellular side of the channel (Ext. in Figure 8), has an intermediate affinity for Ca^2+^ (brown-coloured square); Site 2, in the channel core, has the highest affinity for Ca^2+^ (dark brown); Site 3, close to the intracellular side of the channel (Int.), has the lowest affinity (light brown). At physiological Ca^2+^ concentrations, ions can occupy both Site 1 and Site 3 (state S_a_) given the strong repulsion between Ca^2+^ ions, or a single Ca^2+^ ion can occupy Site 2 (state S_b_). Considering the unidirectional Ca^2+^ flux in physiological conditions, the selectivity filter switches alternatively between the states *S_a_* and *S_b_* (Figure 8). For instance, the transition between *S_a_* and *S_b_* occurs when Ca^2+^ jumps from Site 1 to Site 2 spontaneously (since Site 2 has a higher affinity for Ca^2+^ than Site 1) and/or by ionic repulsion with a third ion that enters on the extracellular side of the filter. Ca^2+^ ion can proceed from Site 2 to Site 3, which has the lowest affinity for Ca^2+^, again for ionic repulsion with another Ca^2+^ ion that rapidly occupies Site 1 for the high Ca^2+^ concentration in the extracellular solution, while the weak binding of Ca^2+^ to Site 3 allows the Ca^2+^ exit toward the low Ca^2+^ concentration of the cytosol. The high-affinity binding of Ca^2+^ to Sites 1 and 2 ensures that Na^+^ and other monovalent cations cannot permeate; if external Ca^2+^ is removed, then monovalent cations can move freely along the channel, giving the observed much larger current (Figure 2B,F, Figure 3C, Figure 6B and Figure 7B). In conclusion, multiple Ca^2+^-binding sites are necessary for permeation, but only Site 2 binds divalent cations with sufficient affinity for blocking monovalent inflow. Surprisingly, and different from K^+^ permeable channels [41], Ca^2+^ seems to be conducted in this mutant channel as a hydrated cation [38,39], on the basis of the structural features of the channel binding sites and on its large estimated functional diameter of 6 Å. Indeed, up to eight water molecules appear to be carried together with a Ca^2+^ ion. This could be actually a way for the selectivity filter to distinguish between Ca^2+^ and Na^+^, because they can be discriminated on their hydrated radius (4.12 and 3.58 Å, respectively) and their charge, but not on their ionic radius (1.00 and 1.17 Å, respectively), as the selectivity filter does for the K^+^ channel [41]. A recent molecular dynamics study based on the crystal structure of the epithelial Ca^2+^ channel TRPV6 confirmed the knock-off mechanism for Ca^2+^ permeation along the Ca^2+^ binding sites of the channel, and confirmed that Na^+^ permeates freely without needing such a mechanism. However, this simulation predicts that in the presence of ions, no water binds to or crosses the pore constriction [40]; therefore, the permeation mechanism of the Ca^2+^ is yet to be fully explained.

The activation time constant (*τ**_on_*) and the fastest time constant of deactivation (*τ**_off1_*) of the maximal cation current were about two and three times faster, respectively, than that of the Ca^2+^ current. This acceleration, which was relatively voltage-independent (Figure 3E) and mainly generated by the acceleration of the activation and deactivation of the L-type channel, did not occur for the longest time constant (*τ**_off2_*) generated by R2 component deactivation. If channel gating is not affected by the kind of ion flowing through the channel, then acceleration of the cation current activation phase vs. that of Ca^2+^ indicates that Ca^2+^ binding to the pore sites (1 and 2; Figure 8) is the rate-limiting step for activation kinetics. This hypothesis predicts that the channel deactivation kinetics would be essentially determined by the gate closure rate, and thus, the deactivation kinetics of the monovalent cation current would be similar to that of the Ca^2+^ current. However, this only occurs for the current flowing through the R2 channel. It is therefore possible that the ion-permeating species affects the L-type (and possibly the R1) channel gating, as demonstrated for the voltage-activated Na^+^ channels [42].

When applied at saturating concentration in 4 mM Ca^2+^, nifedipine blocked the L-type component without any time-dependent effect (Figure 4 and Figure 5): the total current of L + R2 cells was reduced by a constant factor of ~2.4, while in L + R1 + R2 cells, the total current was reduced by a factor of ~3.4. This indicates that the L-type component was somewhat smaller in the cells with two components (~60% of the total current) than in those with all three components (~70%). In 10 nM Ca^2+^, after blocking the L-type component, nifedipine partially blocked the residual current (generated by the R2 channels in the former and by the R1 and R2 channels in the latter) in a voltage- and time-dependent manner. For instance, at −50 mV, this residual current was exponentially reduced (with a time constant of ~6 ms) by 34% in both cases. The blockade was progressively stronger at more hyperpolarised voltages, but peak and plateau were reduced by the same amount (Figure 6D and Figure 7D). When Ca^2+^ is the current carrier, nifedipine appears to block the L-type channel by generating allosteric changes at the channel selectivity filter, impeding Ca^2+^ ion progression in the channel, because it remains blocked at Site 1 (Figure 8, [39]). It is possible that the positive charges that are added when Ca^2+^ is adsorbed to the outside of the membrane stabilise the R2 channel to the point that nifedipine is unable to exert this allosteric change on it. This view is further supported by the fact that the blockade is larger at more negative voltages, i.e., for voltages that simulate more and more the “Ca^2+^ absence”. The interaction of nifedipine with the selectivity filter is also supported by the time-dependent block of the channel, which was evident only for the voltages in which the current was maximal (i.e., between −10 and −60 mV; Figure 6C and Figure 7C), i.e., when it is more difficult to block the large cation influx.

Assuming that in 10 nM Ca^2+^, nifedipine blockade of the R2 channel is negligible at early times of depolarisation, from the results reported in Table 1, it is possible to calculate how much the isolated L current (L), the R2 current (R2), and the R1 + R2 currents (R12) increase in 10 nM Ca^2+^, as follows:(1)$$L=\frac{c-d}{a-b}\;\;\;\;\;\;\;\;R2=\frac{d}{b}\;\;\;\;\;\;\;\;R12=\frac{g}{f}$$
since L = 6.3, R2 = 2.5, and R21 = 2.2 (R1 cannot be discerned from R2 in 10 nM Ca^2+^ for the lack of specific blockers for R1 or R2 components), Ca^2+^ blocks the L channel type more compared to the R1 or R2 ones, since upon removal of external Ca^2+^, the monovalent cation influx (carried mainly by Na^+^) increased more in the former case than in the latter. On the basis of the scheme of Figure 8, the binding Site 2 of the L channel has a higher affinity for Ca^2+^ than that of the R1 or R2 channels.

## 4. Materials and Methods

### 4.1. Animal and Solutions

The experiments were performed on frogs (*Rana esculenta*, 25–30 g body weight) as previously described [2,33,43]. Animal experiments were performed in compliance with the European Communities Council Directive for animal use in science (86/609/EEC) and approved by the ethical committee of the University of Ferrara named “Comitato Etico di Ateneo per la Sperimentazione Animale” (C.E.A.S.A.; authorization number 718/2016-PR, March 2016)”. Briefly, the animals were anaesthetised by immersion in a tricaine methane sulphonate solution (1 g/L in water) and then decapitated. The six ampullae were isolated from the two labyrinths in the dissection solution (composition, in mM: 120 NaCl, 2.5 KCl, 0.5 EGTA, 5 HEPES, 20 sucrose, 3 glucose; pH ≈ 7.2, osmolality ≈ 260 mOsmol/kg) and then transferred into the experimental chamber containing an extracellular solution for patch-clamp recording (composition, in mM: 100 NaCl, 6 CsCl, 20 TEACl, 4 CaCl_2_, 10 HEPES, 6 glucose; pH ≈ 7.2, ≈260 mOsmol/kg). The hair cells were mechanically dissociated from the ampullae by gently scraping the epithelium with fine forceps. The recorded cells were subjected to the external control (composition, in mM: 100 NaCl, 6 CsCl, 20 TEACl, 4 CaCl_2_, 10 HEPES; pH ≈ 7.2, ≈260 mOsmol/kg) and test solutions in rapid succession (typically <50 ms). This was performed by using a computer to move a multi-barrelled perfusion pipette horizontally in front of the recorded cell. The low Ca^2+^ solution and the choline solutions were the same as the control solution, but the former was nominally Ca^2+^-free and contained 1 mM Cs-BAPTA and the latter had 100 mM NaCl substituted with 100 mM of choline chloride. The WinMax program (release 2.1, freely distributed by Purdue University [44]) was used to estimate the resulting free Ca^2+^ of the low Ca^2+^ solution, which proved to be approximately 10 nM, considering the Ca^2+^ contamination of water (estimated at ~5 µM by atomic absorption spectroscopy) and of the salts used to prepare the solutions. Calciseptine (2 µM), calcicludine (20–100 nM), TTX (1 μM), and nifedipine (10 µM) were added to the perfusion solutions after being dissolved in dimethyl sulfoxide. In the imaging experiments, the solutions were focally pressure-applied through a pipette using a gated PicoPump (PV800, World Precision Instruments, Sarasota, FL, USA; Figure 1) to keep the imaged cell from moving under the perfusion flow, which was particularly violent when using the multi-barrelled pipette. The perfusion solution was removed by a peristaltic pump (Masterflex 7533-20, Cole-Parmer, Vernon, IL, USA), which also served to keep the external solution circulating within the recording chamber. The perforated-patch configuration was obtained by adding amphotericin B (240–300 μg/mL) to the pipette solution (composition, in mM: 90 CsCl, 20 TEACl, 2 MgCl_2_, 1 ATP (K^+^ salt), 0.1 GTP (Na^+^ salt), 10 HEPES, 5 EGTA; pH ≈ 7.2, ≈235 mOsmol/kg). The dye used to perform fluorescence imaging of intracellular Ca^2+^ was Oregon green, dissolved in the intracellular solution used for the whole-cell experiment at a 100 µM concentration. This solution also contained 0.5 mM EGTA to chelate residual Ca^2+^ of distilled water and the Ca^2+^ released from the glassware used for solution preparation as well as from the patch pipette itself. All salts, buffers, and solvents were purchased from Sigma Chemical Co. (St. Louis, MO, USA) except for calcicludine and calciseptine (Alomone Labs, Jerusalem, Israel), Oregon green (Molecular Probes, Eugene, OR, USA), and Cs-BAPTA, which was synthesised by our colleagues at the Department of Physics, University of Parma, Italy.

### 4.2. Patch Clamp Recording and Fast Solution Application

Cells were recorded at room temperature (20–22 °C) using the whole-cell or perforated-patch technique and a patch amplifier (EPC7, Harvard Bioscience, Holliston, MA, USA or Axopatch 200B, Molecular Devices, San Jose, CA, USA). The linear leak current was routinely subtracted from all recordings; the capacitive transients were compensated or their digital points were removed from the recordings, since the artefacts were significantly faster than the current onset and offset. The speed of the voltage clamp was adequate to detect the rapid activation and deactivation kinetics of the currents, especially the ones in 10 nM Ca^2+^ (Table 1). These recordings were not considered when the access resistance was larger than 12 MΩ (see Section 2 Results). Since the average cell capacitance was about 7 pF, the speed of the voltage clamp was about 85 µs, more than two times faster than the fastest activation kinetics ever recorded in 10 nM Ca^2+^ (200 µs). Cells were viewed on a TV monitor connected to a contrast-enhanced video camera (IRVX440G, T.I.L.L. Photonics, Kaufbeuren, Germany) coupled to an inverted microscope (IMT-2, Olympus, Tokyo, Japan or TE 300, Nikon, Tokyo, Japan) equipped with a 40X Hoffman modulation contrast objective (Olympus). To facilitate gigaseal formation in the perforated-patch experiments, an amphotericin-containing solution was used to backfill the electrode while the pipette tip was filled with an amphotericin-free solution. The test stimuli consisted of 50 or 160 ms steps from holding potential (*V_h_*, which was −70 mV throughout the recording but before each depolarising stimuli, which was −110 mV) to −20 or −30 mV. The current vs. voltage curves (I–Vs) were recorded starting from *V_h_* and proceeding with 50 ms steps to −20 mV or to +40 mV in 10 mV increments (2 s time interval between steps); there were no differences between the current waveforms elicited by the same depolarising step whether *V_h_* was −70 or −110 mV. Just before and after performing each protocol, leak resistance was measured with a −10 mV step starting from *V_h_* and lasting 15 ms. Data were acquired using a computer interface (Digidata 1322A, Molecular Devices, San Jose, CA, USA) and its managing software (pClamp 8.1, Molecular Devices) running on a host PC. The recordings were filtered at 10 kHz and acquired at 40 kHz via an eight-pole Butterworth filter (LPBS-48DG, NPI Electronic, Tamm, Germany).

### 4.3. Fast Fluorescence Imaging and Data Analysis

Fluorescence imaging of intracellular Ca^2+^ was performed as previously described [33]. Briefly, cells were loaded through the patch pipette with Oregon green (100 µM) dissolved in the pipette solution; recordings were taken within 3 min after attaining whole-cell configuration to allow adequate dye loading. The dye was excited at 494 nm using an interference filter and a long-pass dichroic mirror (D480/30x and 505DCLP, Chroma Technology Corporation, Brattleboro, VT, USA); fluorescence emission was selected at 535 nm using a second filter set (XF23, Omega Optical, Brattleboro, VT, USA) and collected with a x60 objective (0.9 N.A., LUMPlanFl W.I., Olympus, Tokyo, Japan) to form fluorescence images on a fast (15 MHz readout rate) and cooled CCD sensor (IA-D1, DALSA, Waterloo, ON, Canada). The sensor’s output was digitised at 12 bit/pixel by customised electronics to produce 128 × 128 pixel images (resolution ~0.5 µm/pixel), recorded in real time to the RAM of a host PC. The typical inter-frame interval for these recordings was 4.1 ms (frame rate, 244 Hz), close to the relaxation time for the dye–Ca^2+^ binding reaction (approximately 5 ms at room temperature). Data were analysed off-line using a commercial software package (MATLAB 5.3, MathWorks, Natick, MA, USA). In order to correct fluorescence measurements for possible dye concentration and/or optical path length anisotropy, fluorescence signals were computed, for each image pixel, as ratios:$$\frac{\Delta F}{F_{0}}=\frac{F\left(t\right)-F\left(0\right)}{F\left(0\right)}$$
where *t* is time, *F(t)* is fluorescence following a stimulus that raises the Ca^2+^ within the cell, and *F*(0) is the pre-stimulus fluorescence computed by averaging 10–20 images. Both *F*(*t*) and *F*(0) were corrected for mean background fluorescence computed from a 20 × 20 pixel rectangle devoid of obvious cellular structures. The fluorescence ratio magnitude was smoothed with a two-dimensional 3 × 3 median filter and encoded by 8-bit look-up tables to produce 256 pseudo-colour indexed images (shown in Figure 1 inset).

Fittings and statistical analyses were performed with Clampfit (pClamp 8.1 package); figures were prepared using the commercial plotting program Sigmaplot (Version 8.0, Jandel Scientific, San Rafael, CA, USA). Throughout the paper, values in both texts and figures are given as *mean* ± *s.e.m*.; *r* indicates the correlation coefficient.

## Figures and Tables

**Figure 1 ijms-23-03786-f001:**
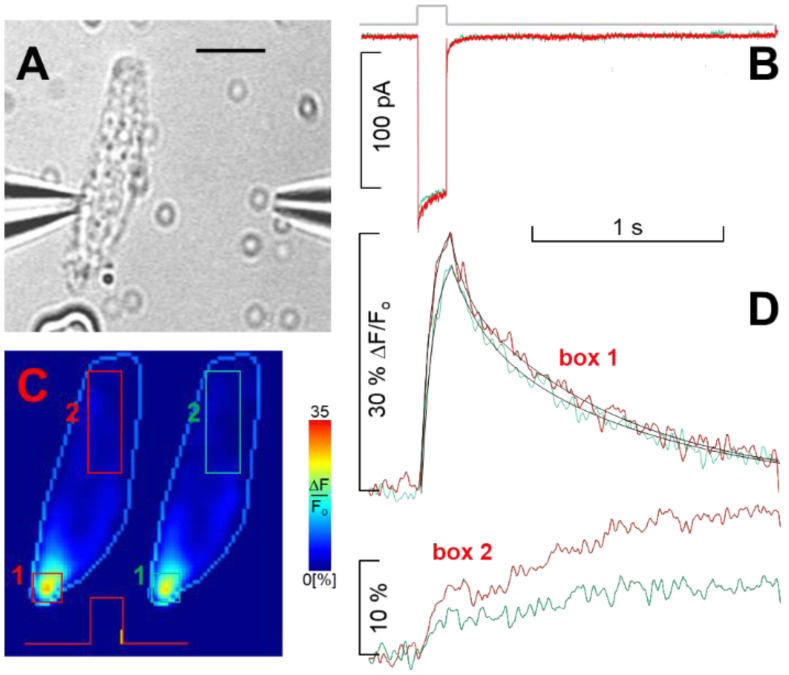
Ca^2+^ channel selectivity in the presence of millimolar extracellular Ca^2+^ and concurrent Ca^2+^ dynamics. (**A**) Bright-field video micrograph of a patched cell (scale bar, 10 µm), showing the location of the puff pipette used for drug application; choline solution ejection started 1 s prior to the onset of the recording and was maintained throughout; (**B**) whole-cell current in the control (red trace) during choline perfusion (green trace) in response to a 160 ms depolarisation to −30 mV from *V_h_*, whose timing is indicated by the thick grey trace, which applies to panel (**D**) as well. (**C**) Two fluorescence ratio (Δ*F*/*F*_o_) pseudo-colour images of the cell in (**A**), encoded according to the colour scale bar on the right (see Section 4 Methods), acquired just at the offset of the voltage pulse (i.e., when intracellular Ca^2+^ is maximal, indicated with the vertical yellow mark superimposed on the voltage command) in the control (left) and during choline application (right); the light blue contour line delineates the cell boundary, enlarged 1.2-fold compared to (**A**) for clarity. (**D**) Time course and spatial distribution of percent Ca^2+^ fluorescence changes computed by averaging pixel signals within the 4 boxes shown (**C**) in the control (red numbers and red traces) and during choline perfusion (green). The black lines in the upper panel are non-linear fits to the rising phase (single exponential, time constant 52 ms for both control and choline perfusion) and to the falling phase (double exponential, time constants *τ*_1_ = 80 ms and *τ*_2_ = 850 ms in the control, *τ*_1_ = 160 ms and *τ*_2_ = 1010 ms in choline) of the fluorescence signals computed in box 1. The lower panel shows the dynamics of Ca^2+^ diffusion from the entry site at the basal pole toward the cell apex (box 2) in the control (red number and trace) and during choline perfusion (green). The same time base applies to both electrophysiological and fluorescence records.

**Figure 2 ijms-23-03786-f002:**
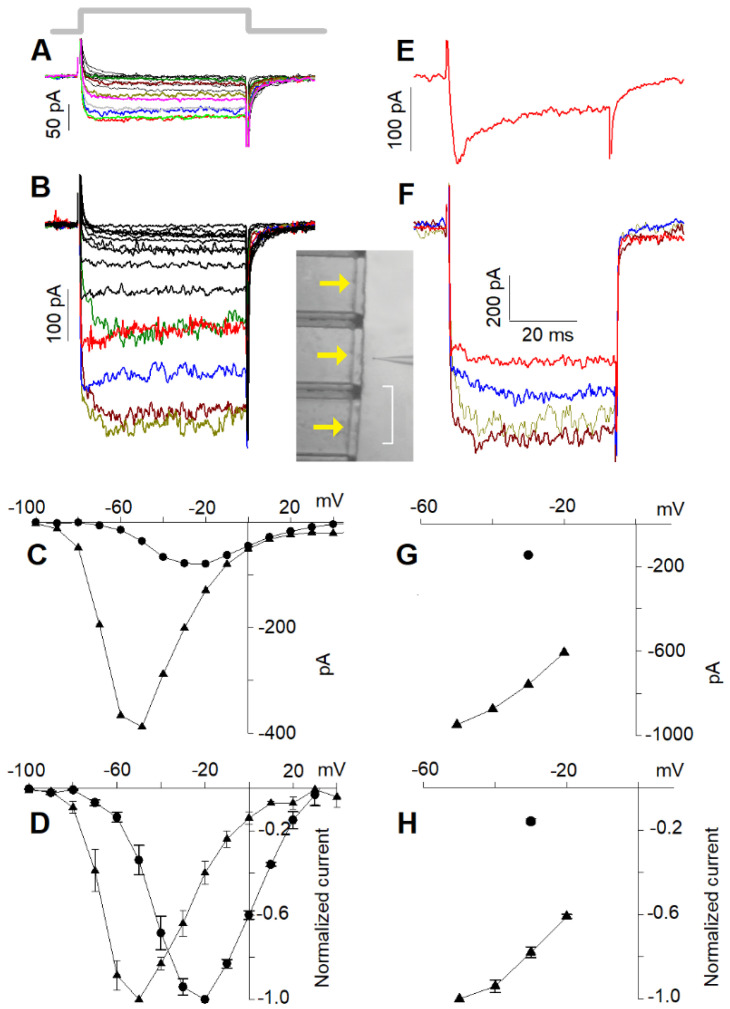
Voltage dependence of the current flowing through the Ca^2+^ channels in 4 mM and 10 nM external Ca^2+^. (**A**) Ca^2+^ currents evoked by the I-V protocol (see Section 4 Methods) in 4 mM external Ca^2+^ in a cell displaying the L and R2 current components (all average values of maximal current amplitudes are reported in Table 1); thick grey line indicates the timing of the 50 ms depolarisations that applies to panel (**B**,**E**,**F**) as well. (**B**) Monovalent cation currents evoked by the I–V protocol of the cell in (**A**) in 10 nM external Ca^2+^. Solution change was attained as described in Methods and is shown in the bright-field video micrograph of the inset (scale bar is 500 µm); yellow arrows indicate the direction of perfusion solution fluxes. (**C**) Steady-state I–V relationships obtained from the tracings in (**A**) (circles) and (**B**) (triangles). (**D**) Same as (**C**); both I–Vs are averaged on *n* = 21 cells normalised to their maximal current in 4 mM Ca^2+^ (circles) and in 10 nM Ca^2+^ (triangles). (**E**) Ca^2+^ current recorded in perforated patch mode evoked by a single depolarising step to −30 mV from *V_h_* in a cell displaying the L, R1, and R2 components. (**F**) Monovalent cation currents of the cell in E in 10 nM Ca^2+^ elicited by four depolarising steps from *V_h_* up to −20 mV in 10 mV increments starting from −50 mV. (**G**) Steady-state I–V relationships obtained from the tracings in (**F**) (triangles); the circle represents the peak Ca^2+^ current of (**E**). (**H**) Same as (**G**); the I–V curve (triangles) and the peak Ca^2+^ current (circle) are averaged on *n* = 9 cells and both normalised to the maximal current in 10 nM Ca^2+^. Some current traces in (**A**,**B**,**F**) are coloured to better identify them (−70 mV, dark green; −60 mV, dark red; −50 mV, dark yellow; −40 mV, blue; −30 mV, red; −20 mV, green; −10 mV, grey; 0 mV, pink).

**Figure 3 ijms-23-03786-f003:**
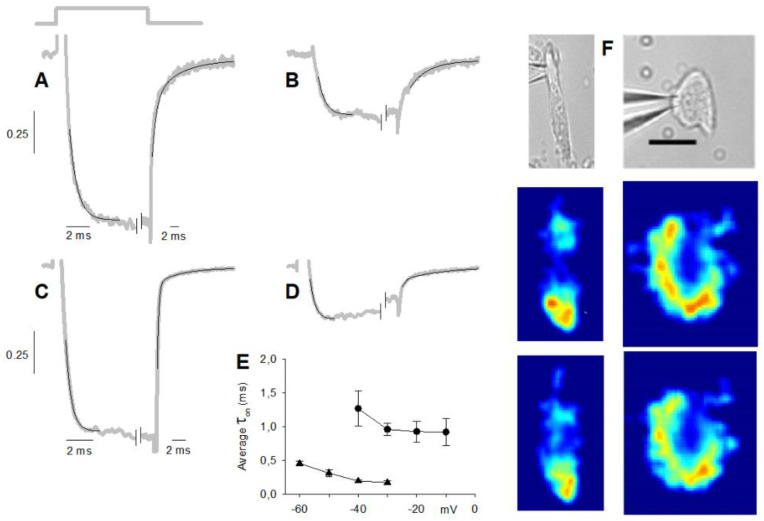
Exemplified exponential fittings (thin black lines) to normalised average current tracings (thick grey lines) of five cells displaying the L and R2 components. (**A**) Fittings to the activation and deactivation kinetics of the average maximal current in 4 mM Ca^2+^; thick grey solid lines indicate the timing of the 50 ms depolarisations that applies to panel (**B**–**D**) as well; note the different time scale than that of panel (**C**). (**B**) Fittings to the average maximal Ca^2+^ current in the presence of 10 μM nifedipine. (**C**) Fittings to the average maximal cation current recorded in 10 nM external Ca^2+^. (**D**) Fittings to the average maximal cation current in the presence of 10 μM nifedipine. Each one of the five cells averaged was subjected to the following fast solution exchange protocol (using the multi-barrelled pipette of Figure 2): 4 mM Ca^2+^, 4 mM Ca^2+^+nifedipine, 4 mM Ca^2+^, 10 nM Ca^2+^, 10 nM Ca^2+^ + nifedipine, and finally 4 mM Ca^2+^; the three recordings in 4 mM Ca^2+^ were almost identical; depolarisation was to −20 mV in (**A**,**B**) and to −50 mV in (**C**,**D**); the horizontal and vertical scales of (**A**) apply to (**B**), the scales of (**C**) apply to (**D**); currents in nifedipine (**B**,**D**) are normalised to their own control ((**A**,**C**), respectively), whose maximal amplitude was considered 1. (**E**) Activation kinetics (*τ**_on_*) of the cells averaged in panel (**A**) (filled circles) and in panel (**C**) (filled triangles) at various voltages. (**F**) Top two images, bright-field video micrograph of two patched cell (scale bar, 10 µm; upper panels); middle two images, Δ*F*/*F*_o_ pseudo-colour images corresponding to the two cells on top, encoded according to the colour-scale bar of Figure 1 (maximal fluorescence was 25% for the cell on the left and 20% for the cell on the right), acquired just at the offset of a 160 ms depolarisation to −30 mV (as shown in Figure 2C) in the control; lower two images are as the middle images but during nifedipine application with the puff pipette shown in Figure 1A; all the fluorescence images are enlarged 1.5-fold compared to the bright-field video micrograph for clarity. Numbers identify the same hotpots in the pseudo-colour images.

**Figure 4 ijms-23-03786-f004:**
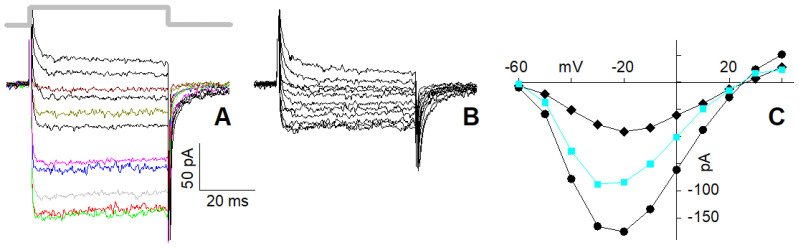
Effect of nifedipine on the L + R2 Ca^2+^ current elicited by depolarisations in 4 mM Ca^2+^. (**A**) Ca^2+^ currents evoked by the I–V protocol (whose timing is indicated by the thick grey line that also applies to panel (**B**)) in a cell displaying the L and R2 components; scales of (**A**) also apply to (**B**). (**B**) Effect of 10 µM nifedipine on the currents of the cell in (**A**). (**C**) Steady-state I–V relationships obtained from the tracing in (**A**) (circles) and (**B**) (diamonds); the difference between the two I–Vs is shown in turquoise (squares). Some current traces in (**A**) are coloured with the same code of Figure 2 to better identify them; traces in (**B**) are not coloured because traces were too close.

**Figure 5 ijms-23-03786-f005:**
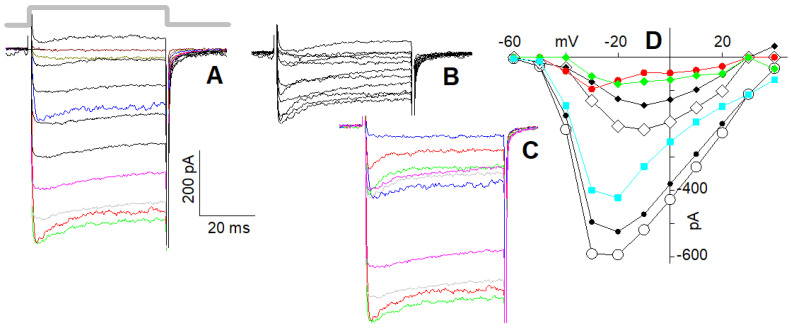
Effect of nifedipine on the L + R1 + R2 Ca^2+^ current elicited by depolarisations in 4 mM Ca^2+^. (**A**) Ca^2+^ currents evoked by the I–V protocol (whose timing is indicated by the thick grey line that applies to panel (**B**,**C**) as well) in a cell (recorded in perforated patch) displaying L, R1, and R2 components (inactivation time constant: 10.8 ms at −30 mV); scales of (**A**) apply to (**B**,**C**) as well. (**B**) Effect of 10 µM nifedipine on the currents of the cell in (**A**). (**C**) Selected traces to the same depolarisation in the control (**A**) and in nifedipine (**B**) (having smaller amplitude of (**A**) traces) are shown together to better compare them. (**D**) I–V relationships of the peak and plateau currents of (**A**) (open circles and filled circles, respectively, i.e., in the control), of the peak and plateau currents of (**B**) (open and filled diamonds, i.e., in nifedipine); plot of the differences between the open and the filled circles (red-filled circles), between the open and the filled diamonds (green-filled circles), and between the filled circles and filled diamonds (turquoise-filled squares). Some current traces in (**A**) and all the traces of (**C**) are coloured with the same code of Figure 2 to better identify them; traces in (**B**) are not coloured because traces were too close.

**Figure 6 ijms-23-03786-f006:**
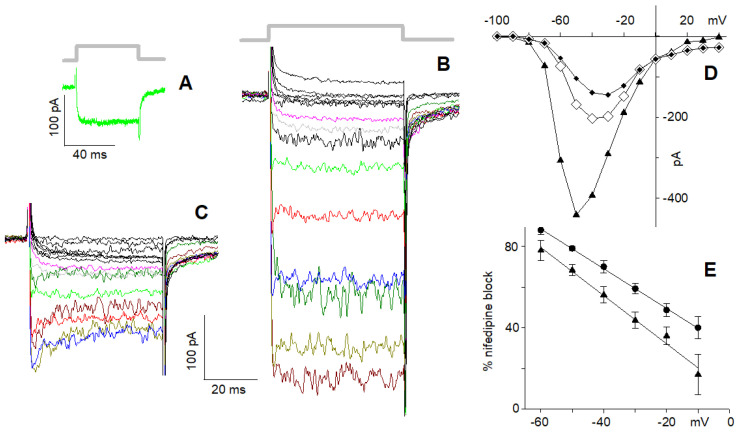
Effect of nifedipine on the L + R2 cation current in 10 nM Ca^2+^. (**A**) Ca^2+^ current evoked by a −20 mV depolarising step (whose timing is indicated by the thick grey line) in 4 mM Ca^2+^ in a cell possessing the L and the R2 components. (**B**) Cation currents in 10 nM Ca^2+^ elicited in the same cell by the I–V protocol (whose timing is indicated by the thick grey line, which also applies to panel (**C**)). (**C**) Effect of 10 µM nifedipine on the currents in (**B**); time constant of the cation current decay at −50 mV was 6.1 ms. (**D**) Steady-state I–V relationships obtained from the tracings in (**B**) (triangles) and from the peak and the plateau currents in (**C**) (open and filled diamonds, respectively); (**E**) Average voltage-dependent fractional block operated by nifedipine on the peak (triangles; *r* = 0.99, slope = −1.28% per mV, *n* = 8) and on the plateau (circles; *r =* 0.99, slope = −0.97% per mV, *n* = 8) of the current in (**C**), calculated using the formula: (1 − *a*/*b*)·100, where *a* is the peak or the plateau amplitude of current in nifedipine (**C**) at a certain voltage, and *b* the amplitude (that is always a plateau) in the control at the same voltage ((**B**); i.e., 100% block occurs if the current amplitude in nifedipine is zero). Some current traces in (**B**,**C**) are coloured with the same code of Figure 2.

**Figure 7 ijms-23-03786-f007:**
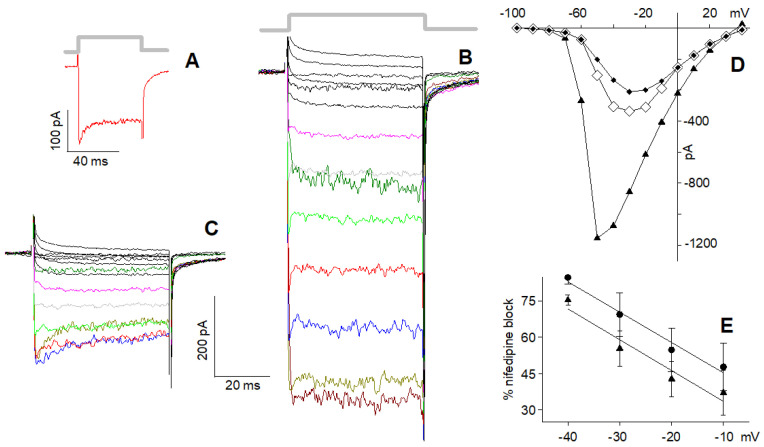
Effect of nifedipine on the L + R1 + R2 cation current in 10 nM Ca^2+^. (**A**) Ca^2+^ current evoked by a −30 mV depolarising step (whose timing is indicated by the thick grey line) in 4 mM Ca^2+^ in a cell (recorded in perforated patch) possessing the L, the R1, and the R2 components. (**B**) Cation currents evoked in the same cell (that was exceptionally stable) by the I–V protocol in 10 nM Ca^2+^ (whose timing is indicated by the thick grey line, which also applies to panel (**C**)). (**C**) Effect of 10 µM nifedipine on the currents in (**B**); (**D**) steady-state I–V relationships obtained from the tracings in (**B**) (triangles) and from the peak (open diamonds) and plateau (filled diamonds) currents in (**C**). (**E**) Average voltage-dependent fractional block operated by nifedipine on the peak (triangles; *r =* 0.97, slope = −1.28% per mV, *n =* 6) and on the plateau (circles; *r =* 0.98, slope = −1.26 % per mV, *n =* 6) of the currents in (**C**). Some current traces in (**B**,**C**) are coloured with the same code of Figure 2.

**Figure 8 ijms-23-03786-f008:**
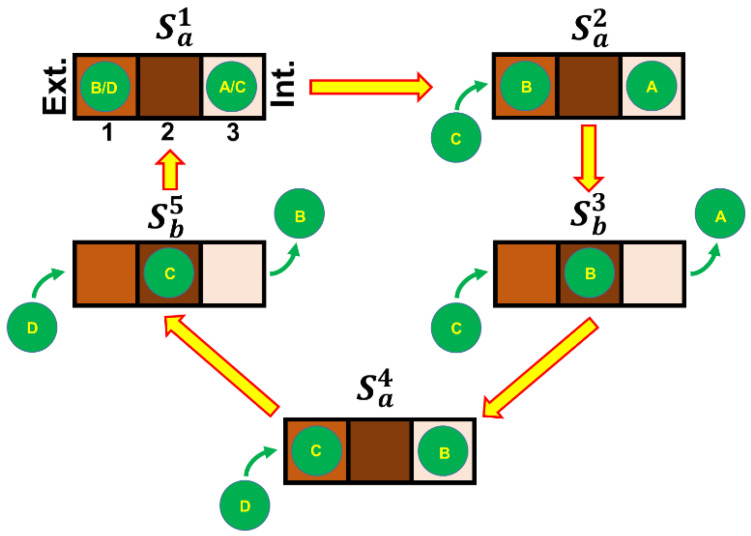
Five “screenshots” of unidirectional Ca^2+^ permeation along the channel selectivity filter. The sequence starts from the screenshot 1 ($S_{a}^{1}$), outlining the channel selectivity filter, that, for the sake of simplicity, depicts it in the *S_a_* state, i.e., with a Ca^2+^ ion (green circle, named (**B**,**D**)) bound in Site 1 (facing the extracellular side, Ext.) and a second Ca^2+^ ion ((**A**,**C**)) bound in Site 3 (Int.). The sequence proceeds with a Ca^2+^ ion (**C**) reaching the extracellular channel vestibule ($S_{a}^{2}$ ), and the electrostatic repulsion with ion (**B**) and the higher affinity of Site 2 move ion (**B**) to Site 2, which in turn pushes ion (**A**) to move into the cytosol (again due to electrostatic repulsion between (**A**,**B**)), returning the selectivity filter to state *S_b_* ($S_{b}^{3}$ ). The binding of ion (**C**) to Site 1 pushes ion (**B**) toward Site 3, switching the filter in state *S_a_* ($S_{a}^{4}$ ). A fourth ion (**D**) reaching the external channel vestibule pushes the ion (**C**) to Site 2, which in turn pushes ion (**B**) to exit the channel, switching the filter in state *S_b_* ($S_{b}^{5}$ ). Finally, ion (**D**) binds to Site 1, pushes ion (**C**) to Site 3, the filter returns to state S_a_, and the cycle repeats.

**Table 1 ijms-23-03786-t001:** Waveform parameters of Ca^2+^ and cation currents in the control and in 10 µM nifedipine.

	4 mM Ca^2+^		10 nM Ca^2+^		4 mM Ca^2+^		10 nM Ca^2+^	
	L + R2 (n = 21)	R2 * (n = 11)	L + R2 (n = 21)	R2 * (n = 8)	L + R1 + R2 (n = 9)	R1 + R2 * (n = 6)	L + R1 + R2 (n = 9)	R1 + R2 * (n = 6)
*τ**_on_* (ms)	0.70 ± 0.06	0.75 ± 0.11	0.36 ± 0.02	0.38 ± 0.03	0.39 ± 0.12	0.40 ± 0.04	0.37 ± 0.04	0.28 ± 0.04
*τ**_off1_* (ms)	1.18 ± 0.16		0.36 ± 0.02		0.89 ± 0.19		0.29 ± 0.02	0.32 ± 0.08
*τ**_off2_* (ms)	9.1 ± 1.1	5.8 ± 0.5	12.0 ± 1.8	6.3 ± 1.0	10.6 ± 1.0	6.6 ± 1.6	11.6 ± 1.9	5.8 ± 1.7
Peak (pA)				156 ± 14	421 ± 80	144 ± 31		317 ± 71
Plateau (pA)	95.8 ± 7.8 ^a^	39.4 ± 4.1 ^b^	457 ± 36 ^c^	100 ± 8.0 ^d^	359 ± 66 ^e^	95 ± 22 ^f^	1250 ± 140	207 ± 62 ^g^

* The L component was blocked with 10 µM nifedipine; ^a−g^ indicate the numbers to be placed in the Equation (1) (i.e.: 95.8 is *a*, 39.4 is *b*, etc.).

## Data Availability

The data presented in this study are available on request.

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
