# Peer review of "Cation Permeability of Voltage-Gated Hair Cell Ca2+ Channels of the Vertebrate Labyrinth"

_ijms, 2022, doi:10.3390/ijms23073786_

Round 1

Reviewer 1 Report

Marta and Giorgio have presented exciting electrophysiological confirmation of Na+ conductance by the calcium selective channels VGCCs present on sensory cells, whech explosed to nanomolar conc of extracellular Ca. They also dissect out the contribution of L-type, P/Q/N-type and R-type channels in this process by using the pharamacological inhibitors of the channels. The data presented is interesting. To improve it further and to make it easier for the readers to understand the data, I advice the authors to put labels for the line traces on the figures themselves. The figure legends are overcrowded, thus making it difficult to understand the findings going to and forth each figure and its legend.

Besides, the text sometimes mentions figure numbers not in sequence, for example Figure 4 is mentioned in the introduction, Figure 4B is the first figure mentioned in results. Fig. 3 is mentioned between fig 1 and fig 2.

The text is largely exhaustive. Please try to avoid repeating some data details that are present in figure legends and in main text.

The title of Section 2.4.1 is imcoplete.

Specificity of the drugs for Cav1,2,3 need to to be shown in HEK293 cells transfected with these channels, one at a time. Alternatively suitable reference should be prodived in the introduction section.

Author Response

Marta and Giorgio have presented exciting electrophysiological confirmation of Na+ conductance by the calcium selective channels VGCCs present on sensory cells, whech explosed to nanomolar conc of extracellular Ca. They also dissect out the contribution of L-type, P/Q/N-type and R-type channels in this process by using the pharamacological inhibitors of the channels. The data presented is interesting. To improve it further and to make it easier for the readers to understand the data, I advice the authors to put labels for the line traces on the figures themselves. The figure legends are overcrowded, thus making it difficult to understand the findings going to and forth each figure and its legend.

ANSWER: to make it easier for the readers to understand the data, and to “lighten” the overcrowded figure legend, we have splitted Figure 4 and Figure 5 in two (so to have four figures instead of two). We have therefore made four figures, four figure legends and change the text accordingly, but we believe this improved very much the overall paper readability. 

Besides, the text sometimes mentions figure numbers not in sequence, for example Figure 4 is mentioned in the introduction, Figure 4B is the first figure mentioned in results. Fig. 3 is mentioned between fig 1 and fig 2.

ANSWER: As suggested by the reviewer, we have corrected the awkward figure mentions, and we have changed it with “see Results” or “see below” where appropriate

The text is largely exhaustive. Please try to avoid repeating some data details that are present in figure legends and in main text.

ANSWER: As suggested, we have “lightened” the figure captions by removing from Figure 2, and from the new Figure 4, 5, 6, and 7, the data whose average values are reported in Table 1

The title of Section 2.4.1 is imcoplete.

ANSWER: A suggested, we have changed the title in “I-Vs of L+R2 and of L+R1+R2 currents in 4 mM Ca2+” We have also changed the title of Section 2.4 in: “I-Vs of L+R2 and of L+R1+R2 currents in 4 mM Ca2+

Specificity of the drugs for Cav1,2,3 need to to be shown in HEK293 cells transfected with these channels, one at a time. Alternatively suitable reference should be prodived in the introduction section.

ANSWER: We agree with the reviewer, the Introduction is lacking the discussion of the specific drugs against the different calcium channels. We tried to insert a small chapter discussing this topic, but since there are very extensive (and excellent) reviews on this subject, we have added a quite complete review to the reference list and we have added the following chapter in the Introduction:

Ca2+ channels are classified according to the properties of the current they carry: the L-type currents originate from Cav1 channel family, the P/Q-, N-, and R-type currents arise from Cav2 one, and T-type currents are mediated by Cav3 one. This classification is the outcome of an immense interdisciplinary research lasting more than fifty years, by studying primarily the effects of specific drugs on these channels in ex vivo tissues, in vivo, and expressed (native or mutated) in cell lines [doi: 10.1002/wmts.102].

Reviewer 2 Report

The manuscript provided by Marta Martini and Giorgio Rispoli demonstrates the aspects of L- and R-type voltage-gated calcium channels (VGCC) permeability in labyrinth hair cells. The study demonstrates the ability of Na+ penetrates through the VGCC at low extracellular Ca2+ concentration. Of course, these conditions are non-physiological, but using this model interesting data about biophysical and pharmacological properties of VGCC was obtained. I have some concerns regarding data interpretation:

Major points:

  • In addition to TTX-sensitive channels, the presence of TTX-resistant voltage gated Na+ channels have been found in some cell types. It is reported in section 2.2 that monovalent currents are not affected by TTX; however, TTX-resistant currents were shown in hair cells (3389/fnins.2021.733291). Please discuss this issue or provide the experimental data confirming the absence of contribution of TTX-resistant channels.
  • All explanations of the authors rely on the fact that Ca2+ currents are mediated by L- and T-type VGCC. How do the authors exclude the contribution of T-type channels into the Ca2+ currents without experiments with T-type channels blockers such as ML-218?
  • How did the authors chose the concentration of nifedipine (10 μM)? Nifedipine demonstrates lower affinity than isradipine and is used as a ruler at micromolar range. However, non-specific action of this blocker (and other dihydropyridines) was found (https://doi.org/10.3390/ijms221910342; 10.1073/pnas.0936131100). This fact should be considered in the experiments.

Minor points:

  • The authors state that Ca2+ and Na+ ions have the similar diameter (2Å) but do not provide the appropriate reference. Is it diameter of the solvated ion or diameter without hydrated shell?
  • Please, clearly indicate the percentage of cells with L/R2 and L/R1/R2 components of Ca2+
  • Typographical errors have to be corrected.

Author Response

Major points:

In addition to TTX-sensitive channels, the presence of TTX-resistant voltage gated Na+ channels have been found in some cell types. It is reported in section 2.2 that monovalent currents are not affected by TTX; however, TTX-resistant currents were shown in hair cells (3389/fnins.2021.733291). Please discuss this issue or provide the experimental data confirming the absence of contribution of TTX-resistant channels.

ANSWER: This is an important point raised by the reviewer that needs an extensive discussion, that we have included in the paper, that we believe it exhaustively answers to this referee’s concern:

“Indeed, replacing all external 100 mM Na+ with impermeant cation choline, in the presence of invariant 4 mM Ca2+, produced no significant changes in current amplitude and waveform elicited by the depolarization in an hair cell (Figure 1A) that contained all three channel types (Figure 1B). This excludes also the presence of Nav channels, as the TTX insensitive ones that peaks at -30 mV, recently described in human fetal vestibular hair cells [31], otherwise the current amplitude (in response to a -30 mV depolarization; Figure 1B) would be reduced in choline. This experiment excludes also the presence of a TTX-sensitive Nav channel, as the one described in type I and type II auditory hair cells of the chicken (blocked by sub-micromolar concentrations of TTX; Kd≈17 Nm [32]). Another evidence demonstrating the absence of a TTX-sensitive Nav is the almost identical current waveform, elicited by depolarizations in 4 mM or in 10 nM external Ca2+, recorded before, in the presence, and after the application of TTX (1 μM, using the fast perfusion system described in Methods and shown in the video microphotograph of Figure 2).”

All explanations of the authors rely on the fact that Ca2+ currents are mediated by L- and T-type VGCC. How do the authors exclude the contribution of T-type channels into the Ca2+ currents without experiments with T-type channels blockers such as ML-218?

ANSWER: We agree with the reviewer, we cannot exclude that the R1 channel is actually a T-type. We suspected that the R1 channel was a R-type one on the basis of its blockade by mibefradil [10.1016/S0006-3495(00)76681-1], that it has been shown to block the R-type channel [10.1016/s0028-3908(97)00086-5.], and that they inactivate in a Ca2+-dependent manner, like other R-type channels described in neurons [10.1016/s0896-6273(03)00560-9]. However, the paper [10.1016/s0028-3908(97)00086-5.], indicates that mibefradil blocks T-type channels as well. Since the goal of our paper is to study the Na+ permeability in hair cells  Ca2+ channels,  rather than identify their exact type (that would also require many experiments of molecular biology, besides the use of T-type specific drugs), we have added to the Introduction the following paragraph, discussing the possibility that the R1 channel could be indeed a T-type one:

“Moreover, the R1 component, originally identified as an R-type because it was sensitive to the R-type channel blocker mibefradil [2], could be instead a T-type channel, that is mibefradil-sensitive as well [28], and possibly a Cav3.1 one, giving its very fast inactivation kinetics [29]. In conclusion, the channel classification presented here should be considered still not conclusive.”

How did the authors chose the concentration of nifedipine (10 μM)? Nifedipine demonstrates lower affinity than isradipine and is used as a ruler at micromolar range. However, non-specific action of this blocker (and other dihydropyridines) was found (https://doi.org/10.3390/ijms221910342; 10.1073/pnas.0936131100). This fact should be considered in the experiments.

ANSWER: As stated in the original version of the paper (lines 68-72) “nifedipine (or nimodipine) concentrations of 1, 5 and 10 μM gave the same fractional block of the total current (about 60-70%; [2,12]), without affecting the peak phase of the current [2].” This indicates that 10 μM of nifedipine is a concentration that fully blocks the L component, leaving the recordings with the R1 component (inactivating) and the R2 component. We did not use here nifedipine concentrations larger than 10 μM , to avoid to affect the R1 and the R2 components: it is reasonable to suspect that, by increasing the concentration of any Ca2+ antagonist, any Ca2+ channel would be affected before or after (or even any other cationic channel). In conclusion, according to the referee suggestion, we added the following paragraph at the beginning of chapter 2.3:

“Since nifedipine concentrations between 1 and 10 μM gave the same suppression of the Ca2+ current, it was concluded that this concentration range sufficed to fully block the L component of the current. In the following, it was chosen a concentrations of 10 μM to ensure the full block of the L component, but no larger concentrations were employed, since they block other Ca2+ channel types [34] and could therefore affect the R1 and R2 components of the current as well.”

Minor points:

The authors state that Ca2+ and Na+ ions have the similar diameter (2Å) but do not provide the appropriate reference. Is it diameter of the solvated ion or diameter without hydrated shell?

ANSWER: This is actually another important point raised by the reviewer, that we have overlooked in the paper, that is now discussed as follows:

“Voltage gated Ca2+ channels are extremely selective for Ca2+ ions, since they convey a Ca2+ influx of 106 ions/sec into the cell, despite extracellular Na+ is 70-fold more concentrated than Ca2+ and both ions have nearly identical dimension. Indeed, the Van der Waals, ionic, and hydrated radius of Ca2+ are, respectively, 2.31, 1.00, and 4.12 Å, while the ones of Na+ are: 2.27, 1.17, and 3.58 Å [8], suggesting that the Ca2+ channel selectivity filter could rely on recognizing the hydrated form of Ca2+ to select it over Na+ (see Discussion).”

In the Discussion, we have added this paragraph:

“Surprisingly, and at difference with K+ permeable channels [41], Ca2+ seems to be conducted in this mutant channel as a hydrated cation [38,39], on the basis of the structural features of the channel binding sites and on its large estimated functional diameter of 6 Å. Indeed, up to eight water molecules appear to be carried together with a Ca2+ ion. This could be actually a way for the selectivity filter to distinguish between Ca2+ and Na+, because they can be discriminated on their hydrated radius (4.12 and 3.58 Å, respectively) and their charge, but not on their ionic radius (1.00 and 1.17 Å, respectively), as it does the selectivity filter for the K+ channel [41].”

Please, clearly indicate the percentage of cells with L/R2 and L/R1/R2 components of Ca2+

ANSWER: At the beginning of chapter 2.1. (lines 121-123 of the original paper) it is stated:

“As found previously, approximately 60% of the hair cells recorded in whole-cell mode exhibited a steady current, maximal at -20 mV, whereas the remaining 40% were characterized by an initial peak, followed by an exponential decay (inactivation) to a plateau level.”

Typographical errors have to be corrected.

ANSWER: The paper has been carefully checked for typographical and for any other kind of errors that were present in the first version of the manuscript, we found quite a few of them.

Round 2

Reviewer 2 Report

All my comments have been addressed.